# Collective language creativity as a trade-off between priming and antipriming

**Sergei Monakhov** *

Friedrich Schiller University, Jena, Germany

* sergei.monakhov@uni-jena.de

## Abstract

It is now a matter of scientific consensus that priming, a recency effect of activation in memory, has a significant impact on language users' choice of linguistic means. However, it has long remained unclear how priming effects coexist with the creative aspect of language use, and the importance of the latter has been somewhat downplayed. By introducing the results of two experiments, for English and Russian native speakers, this paper seeks to explain the mechanisms establishing balance of priming and language creativity. In study 1, I discuss the notion of collective language creativity that I understand as a product of two major factors interacting: cognitive priming effects and the unsolicited desire of the discourse participants to be linguistically creative, that is, to say what one wants to say using the words that have not yet been used. In study 2, I explore how priming and antipriming effects work together to produce collective language creativity. By means of cluster analysis and Bayesian network modelling, I show that patterns of repetition for both languages differ drastically depending on whether participants of the experiment had to communicate their messages being or not being able to see what others had written before them.

**Data Availability Statement:** All data are available in the Zenodo repository, https://doi.org/10.5281/zenodo.4739545.

**Funding:** The author(s) received no specific funding for this work.

## Introduction

It is a matter of scientific consensus now that priming, a recency effect of activation in memory, has a significant impact on language users' choice of linguistic means. There is a large body of research indicating that recently activated discourse elements have higher likelihood of being reused in the unfolding situation [1–6]. However, it has long remained unclear how priming effects coexist with the creative aspect of language use, and the importance of the latter was somewhat downplayed.

Starting from Noam Chomsky, who declared creative aspect of language use to be a mystery that the rules and principles of grammar can only shed light on but never give full account of [7: 425], language creativity has been an area of extensive research for decades [8–18]. While Chomsky distinguished between creative aspect of language use and 'true creativity in a higher sense,' discussing also 'rule-changing creativity' as well as the creative use of particular rules and canons (ibid.), a substantial part of the research to follow focused on analysing the choice of words and turns of phrases. The predominant ideas here were those of 'schema refreshing',

**Competing interests:** The authors have declared that no competing interests exist.

'heteroglossia' [19, 20], and 'people's strategic deployment of language resources from across their repertoire' [21].

The theoretical discrepancy should have become obvious here: if speakers' choice of linguistic means is affected by priming, why do they then engage in all sorts of language games whose main point is avoiding repetitiveness? It should have, but to the best of my knowledge, somehow it hasn't. Importantly, whereas priming was and remains one of the core theoretical concepts of modern cognitive linguistics, most of the aforementioned works on creativity came from somewhat marginal (or at least much more specialised) domains: language play, language learning, humour in language, figurative language, narration techniques, etc.

Still, it is clear that people do use language creatively, and hence, both phenomena, which until now have received very different amounts of scientific attention, are equally important. On a more general, conceptual level, this was of course recognised long ago. Among famous maxims of action proposed by Rudi Keller as part of his invisible-hand theory for explaining language change [22: 95–107], one finds maxims of conformity and extravagance (names suggested in [23: 1055]): 'talk like the others talk' and 'talk in such a way that you are noticed', respectively. Narrowing these maxims' original meaning, one could say that priming effects force people to do the first thing, while creativity urges to do the second. But how exactly are they correlated, how do they interact with each other?

Surprising as it may seem, a new insight into the problem of language creativity came from an unexpected venue, namely, from studying discourse practices of internet trolls. Monakhov [24, 25] showed that a number of features inherent in trolls' tweets are grounded in the sociolinguistic limitations of this type of discourse, which, in essence, is an imitation, make-believe game. A troll wants to achieve their goal without being identified as trying to achieve it. In other words, their language attitudes are moulded by a combination of two factors: first, communicating a limited number of messages multiple times (speaking with a purpose); second, faking the diversity of contexts and topics in the process (trying to mask the purpose of speaking).

It means that, though troll writing is usually thought of as being permeated with recurrent messages, its most characteristic trait is an anomalous distribution of repeated words and word pairs. This anomaly is inevitable because a secret task of delivering a target message multiple times without being suspected of such can only be fulfilled by using a limited number of signal words in a wide variety of different contexts. Suppose that a troll has to write a great number of messages using the words *vaccine* and *death* next to each other. It is not possible to simply continuously repeat the same tweet because that will lead to the exposure of the troll. Hence, it is necessary to use the target words in a variety of different contexts, including those where they may seem incongruous to most speakers. This, in turn, has consequences for the target words' lexical compatibility: their distribution markedly increases, their neighbours become more numerous, and the co-occurrence links between them and other words become artificially strengthened.

The effect may be likened to the famous ciphering technique described in the short novel by Sir Arthur Conan Doyle *The Gloria Scott* (1894). There, one of the characters received the following grotesque message: 'The supply of game for London is going steadily up. Head-keeper Hudson, we believe, has been now told to receive all orders for fly-paper and for preservation of your hen-pheasant's life.' The key to this riddle was in Sherlock Holmes's hands when he saw that only every third word, beginning with the first, should be taken into account, which gave him 'The game is up. Hudson has told all. Fly for your life.' Now imagine that the author of this warning has to resend it as different messages several hundreds of times. The target words will naturally remain the same, but the filler words will change. For that reason, the target word *game*, for example, will be part of ever changing non-collocational bigrams (like *supply of game* in the citation above), which will result in the number of repetitions of this

word going up and the number of repetitions of bigrams with this word going down. Given the collocational nature of language production, such linguistic behaviour may be considered anomalous.

Building upon this hypothesis, Monakhov proposed a simple and effective algorithm for the identification of troll writing, which was based on calculating the ratio of the proportion of repeated content words among all content words to the proportion of repeated content word pairs among all content word pairs. He found that, regardless of the distribution of topics in tweets and the number of content words within a message, tweets written by trolls were characterised by greater values of this ratio than tweets written by congresspeople and Donald Trump, which were used for comparison. The reason for this was that the denominator always had a higher value in the latter case than in the former, since repeated content word pairs were more frequent in non-troll writing.

Interestingly, Monakhov's algorithm, when tested on 180,000 mobile app reviews, identified as troll-like not only suspicious five-star comments left by users of some dubious programs, but also strictly negative one-star reviews (under review). Of course, it would be very strange to think that one-star reviews are actually not genuine but written for whatever reason by a group of paid authors. There appears to be only two logically consistent ways to account for this phenomenon: either the algorithm for some reason works only with tweets but not with app reviews, or the very communicative situation of posting an app review somehow leads to the emergence of collective 'troll effect'. The first explanation seems implausible: there are no discernible features that set apart, with regard to the numbers of repeated content words and word pairs, topically related tweets, on the one hand, and app reviews, on the other, as two varieties of short internet messages. The second explanation, however, is psychologically credible. One might assume that it is very common for internet commenters to first read what other people have written and only after that share their own opinion.

Having put this hypothesis to test, Monakhov showed that the quantitative measure of collective language creativity can be derived, for any given sample of messages, as the ratio of the proportion of repeated words among all words to the proportion of repeated word pairs among all word pairs. This ratio was found to decrease as the time distance between the elements in the sample increased, proving that interlocutors are more likely to engage in the game of deployment of language resources when a particular communicative space is conceptualised as continuous, which presupposes a very small time distance between the adjacent contributions or (somewhat surprisingly) a very slow rate of new contributions' arrival.

The app reviews are produced in a very special communicative situation that occupies an intermediate position on the monologic-dialogic continuum and therefore makes a perfect setting for studying how pure language creativity is borne as a 'third-type phenomenon', a result of independent efforts of many speech-act participants. To find out whether the observed tendency is truly a factor in human communication in general, it is needed to move from observational data to an experimental setting.

By introducing the results of two specially designed experiments, one for English and one for Russian native speakers, this paper seeks to explain the mechanisms establishing balance of priming and what will be further referred to as antipriming in human communication. The term 'antipriming' is not new in the literature [26, 27]; however, here it is used not in the traditional sense of the 'flip-side' of the priming coin, an associated cost of priming that results in subjects' temporary inability to identify some things that have not been primed. Rather, by antipriming I mean a person's desire to adjust his or her message with regard to what other people have already said within some timeframe that he or she considers relevant. The reason for this adjustment is the unsolicited desire to be linguistically creative, that is, to say what one wants to say using the words that have not yet been used.

The reason to conduct the study in two languages was to see if any effect would be true cross-linguistically. The choice of English and Russian was a matter of convenience sampling and reflects the pilot nature of the current project.

The rest of the paper is structured as follows. In Study 1, I introduce the concept of collective language creativity and describe an algorithm to model it. In Study 2, it is discussed how cognitive priming effects are set off by creative antipriming efforts of the participants in the communication process.

## Study 1. Data and methods

I designed and conducted two experiments in which English and Russian native speakers were asked to write a short (one or two sentences) comment explaining what is going on one picture—a photo of a young man in a tuxedo, standing with a portable sewing machine in hands in front of a truck that has slid into a ditch. I chose it for two reasons: first, it has some kind of mystery to it and is interpretation-inducing, which allowed me to mask the true purpose of study under the pretence that I am interested in elucidation; second, it has several knots and possible hermeneutical lines [28], which allowed me to mimic the actual multiplicity of stories that is somewhat akin to the communicative situation of many customers describing their personal experiences with one and the same app.

Two crowdsourcing platforms were used for the experiment: Amazon Mechanical Turk (https://www.mturk.com) for English speakers and Yandex Toloka (https://toloka.yandex.ru) for Russian speakers. Participants were randomly assigned to one of the two conditions: 1) in the first condition, they had to communicate their message without being able to see what anyone else has written; 2) in the second condition, they had the possibility (but no necessity) of reading what others have written before them. On the platforms, two special task templates were programmed to match two experimental conditions. The first template contained a link to the picture that participants were asked to describe and an input field where they were supposed to type in their comments. The second template contained a link to the specially created webpage, and two input fields. In this condition, participants were supposed to 1) go to the webpage with the picture on top of it and two-column and 250-row table below, 2) look at the picture, scroll down to the first empty row of the table, type in the first column of this row next consecutive number, type in the second column of this row their comment, 3) copy their number and comment, return back to the task page on the respective platform, paste their number and comment from the table into respective input fields.

After creating task templates, I assembled two pools of users registered on the platforms who met the only criterion of being a native speaker of respective language. People were distributed among the pools randomly, so that they knew the general task but were not aware of which experimental condition they will be assigned to. The instructions for the participants of the experiment were, apart from describing the formal ways to proceed, identical and written so as not to reveal the true purpose of study. For each task, a time limit of 10 minutes was imposed and each following task was distributed only after the previous was closed. No user could see any tasks other than those assigned to their pool and was dismissed from the project immediately after submitting the first assignment, so that no one had the possibility to leave more than one comment. After completing the tasks, each participant was rewarded in the amount of $1.0 USD for their submission.

All the necessary precautions were taken to verify the consistency of results in the second experimental condition by comparing numbers and answers in the tables exported from the project webpages and from the crowdsourcing platforms row by row. Everything matched, no discrepancies were detected, which suggests that the operating procedure was well-planned.

Overall, to the first condition were assigned 200 English native speakers and 192 Russian native speakers; to the second condition were assigned 202 English native speakers and 193 Russian native speakers. All submissions were accepted without any censoring provided that they included no less than two content words.

The data for analysis were collected as follows. For each comment, I obtained its text as a sequence of lemmatised content words. The comments in each of the four groups (two languages times two experimental conditions) were chronologically aligned, from the latest to the newest $\{r_1, \ldots, r_n\}$, where $r_1$ is the first, most dated comment and $n$ is the total number of comments. After that, I divided the data into a number of samples by means of the following procedure: 1) a sampling window of 50 comments was chosen at the initial stage (Monakhov 2020) and the first 50 comments were taken as sample $S_1 = \{r_1, \ldots, r_{50}\}$; 2) at each subsequent step, the size of the sampling window was incremented by one element resulting in the family of sets $F = \{S_2, \ldots, S_{n-50}\}$, where $S_2 = \{r_1, \ldots, r_{50+1}\}$, $S_3 = \{r_1, \ldots, r_{50+2}\}$ and so on; 3) from each set of the family $F$, 50 comments were randomly sampled to make their collective creativity coefficients (CCC) comparable. The latter were obtained for each sample in accordance with the Monakhov's formula:

$$q = \frac{(W - w)/W + \varepsilon}{(P - p)/P + \varepsilon},$$

where $w$ is the number of unique content words in a sample, $W$ is the total number of content words, $p$ is the number of unique content word pairs, $P$ is the total number of content word pairs, and $\varepsilon = 0.001$ is used to make sure that $q$ will not be undefined even for the samples with no repeated words. Word pairs for each message in a sample were constructed disregarding the alphabetic ordering of words so that a four-word message $\{A, B, C, D\}$ gave six word pairs $\{AB, AC, AD, BC, BD, CD\}$.

To better understand the logic behind this between-text measure, it is instructive to consider some simple examples (Table 1). One can have a group of four-lexeme messages that share no common words at all. In this case, the CCC of this group will be one. Conversely, one can have a group of four-lexeme messages where each message is an exact replica of others. In this case, troll coefficient will also be equal to one. Now let's see what happens when we reduce

Table 1. Exemplary calculations of CCC for samples with different numbers of repeated words.

| | Samples | | | | $W$ | $w$ | $P$ | $p$ | $q$ |
|---|---|---|---|---|---|---|---|---|---|
| 1 | Word1 | Word2 | Word3 | Word4 | 12 | 12 | 18 | 18 | 0.001 / 0.001 = 1 |
| | Word5 | Word6 | Word7 | Word8 | | | | | |
| | Word9 | Word10 | Word11 | Word12 | | | | | |
| 2 | Word1 | Word2 | Word3 | Word4 | 12 | 4 | 18 | 6 | 0.667 / 0.667 = 1 |
| | Word1 | Word2 | Word3 | Word4 | | | | | |
| | Word1 | Word2 | Word3 | Word4 | | | | | |
| 3 | Word1 | Word2 | Word3 | Word4 | 12 | 6 | 18 | 12 | 0.501 / 0.334 = 1.49 |
| | Word1 | Word2 | Word3 | Word5 | | | | | |
| | Word1 | Word2 | Word3 | Word6 | | | | | |
| 4 | Word1 | Word2 | Word3 | Word4 | 12 | 8 | 18 | 16 | 0.334 / 0.112 = 2.98 |
| | Word1 | Word2 | Word5 | Word5 | | | | | |
| | Word1 | Word2 | Word7 | Word8 | | | | | |
| 5 | Word1 | Word2 | Word3 | Word4 | 12 | 10 | 18 | 18 | 0.167 / 0.001 = 167.66 |
| | Word1 | Word5 | Word6 | Word7 | | | | | |
| | Word1 | Word8 | Word9 | Word10 | | | | | |

the number of repeated words one by one in the comparable groups of messages. With three repetitions, CCC rises to 1.49; with two repeated words, it rises to 2.98; and with just one repeated word, it blows up to 167.66. Monakhov argued that this extreme latter scenario is more typical for troll writing, while the first two are more characteristic of genuine communication, since words of language tend to have well-established collocates and repeating a word tends to result in repeating a word pair.

I contend that in genuine communication, high values of CCC indicate such discourse scenarios where participants have the possibility and consider it relevant to avoid repetitiveness by adjusting their own messages with regard to what other people have already said or written. The null hypothesis $H_0$ of the experiment is that there will be no significant difference in the distributions of CCCs between two experimental conditions in both languages. The alternative hypothesis $H_1$, given our prior state of knowledge, can be formulated as follows: submissions in the first condition will reveal no linear dependence of CCC on their sequence number (sample index), while in the second condition some sort of linear trend will be clearly identifiable.

## Study 1. Results

The results are plotted in Fig 1 in blue colour. As expected, comments produced in the first experimental condition, when sorted from earliest to latest, reveal no association between CCC and range of sampling in both language (English: $r = 0.05$, $p = 0.54$; Russian: $r = 0.001$, $p = 0.98$), while in the comments produced in the second experimental condition, given the

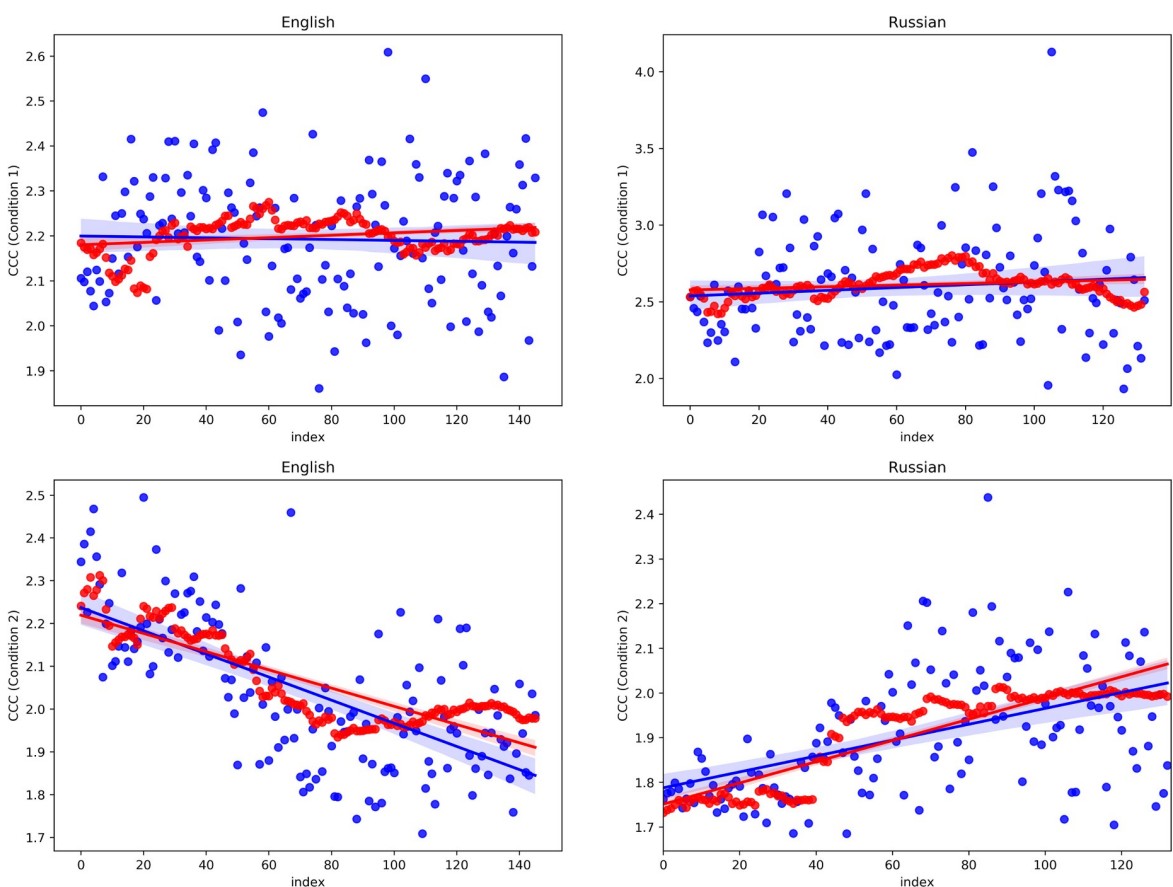

**Fig 1. The distribution of CCCs in English and Russian in two experimental conditions.**

same ordering, a clear linear trend is observed. In English, each increase in sample index predictor leads to a decrease in CCC response ($r$ = -0.66, $p < 0.001$); in Russian, each increase in sample index predictor leads to an increase in CCC response ($r$ = 0.48, $p < 0.001$).

It is important to reiterate what the sample index tells us. The bigger its value, the bigger the range of sampling, that is, the greater is average distance between the comments that appear in this sample. This distance is both chronological and spatial since, due to the experimental design, comments that were written earlier appear higher in the table. In the experimental condition 1, where commenters were not able to see each other's messages, the distance between comments, naturally, could have no bearing on the distribution of repeated words and word pairs. However, in the experimental condition 2, where such possibility existed, the distance between comments influenced this distribution directly, though in opposite ways for two languages. In English, increasing the sparsity of the messages led to their greater independence from one another. In Russian, on the contrary, the positive correlation between CCC and sample indices suggests that the more previous answers are available for participants, the more they try to be linguistically creative.

The larger fluctuations at larger sample indices that can be observed in Fig 1 are undoubtedly explained by the nature of the sampling procedure which implies a weaker effect of random selection on the data when the range is small, than when the range is big. To get a better reflection of the true variability of the CCC as the sample index grows, I calculated the CCC of each sample for a number of randomisations and averaged the results. The number of randomisations was set equal to the sample index, so that, for example, the CCC of the sample $S_2$ was averaged across two randomisations, the CCC of the sample $S_{50}$ across 50 randomisations, etc.

The obtained values are plotted in Fig 1 in red colour. One can see that they mostly lie in the middle of the blue dots' range and closely follow the aforementioned trends in the experimental condition 2 (or reveal absence thereof in the experimental condition 1). The index-dependent variance is greatly reduced and the coefficients of determination for both languages significantly increase: from 0.43 to 0.71 in English and from 0.20 to 0.77 in Russian. Thus, the sampling procedure may be considered reliable.

The difference the two languages reveal in the second experimental condition is unexpected and puzzling, demanding an explanation. The closer investigation of the second row of plots in Fig 1 convinces us that the data patterns here are suggestive of Simpson's paradox [29]: A statistical association that holds for an entire population is reversed in every subpopulation [30]. In other words, in both languages, the data are comprised of a number of small subpopulations of samples, for each of which a negative correlation between CCC and sample index is observable. The only difference is that in English, the mean CCCs of these subpopulations continuously decrease, while in Russian, they increase.

Given observed complexity, one expects to find that spline regression model will fit the data in a better way. As pointed out by various authors, the high flexibility of spline modelling comes at the price of a number of tuning parameters, and the most important among them is the number and spacing of the knots [31]. I approached this problem by identifying as knots those samples where the number of newly introduced words, that is, words which had not appeared in any previous sample was more than three. The results are plotted in Fig 2 (English) and Fig 3 (Russian).

Comparing goodness-of-fit metrics of simple linear and spline regression models for both languages proves that the latter explains in both cases the greater amount of variation, that is, fit the data in a more accurate way (Table 2).

The two patterns show remarkable similarity in a number of aspects. First, the overall structure is similar: 1) initial stationary distribution extending approximately up to the 40th sample

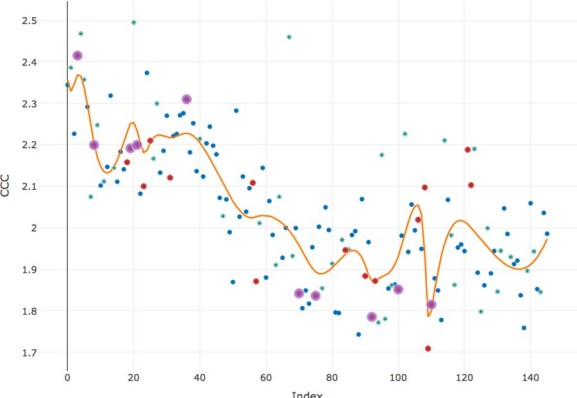

**Fig 2. The distribution of CCC (condition 2) in English modelled by spline regression; red dots indicate samples with more than three new words, purple dots—with more than five.**

(that is, encompassing first 90 comments), 2) intermediate linear, either upward or downward, development, and 3) final stationary distribution that is reached approximately at the 80th sample (that is, after 130 comments). Second, each general trend has a characteristic roller-coaster structure, with peaks and valleys, which can be decomposed into a number of stretches going alternately up and down. The numbers of these stretches, as well as the numbers of knots are close in both languages: 22 knots in Russian and 24 knots in English. The list of similarities may be extended: comparable are the total number of words (472 in English, 453 in Russian), the number of words in the first sample of 50 comments (209 in English, 184 in Russian), the mean rate of new words accumulation (1.75 word per sample in English, 1.97 word per sample in Russian). The question is: why, given all these shared features, the resulting pictures are so different?

## Study 1. Discussion

In terms of graph theory, one can represent a given collection of comments as a network, in which all attested unique words will be nodes, and edges will connect all pairs of words appearing together at least in one comment. Then, a clique of words can be defined as a subset of the nodes, such that every two distinct nodes are adjacent. A maximal clique, that is, a clique that

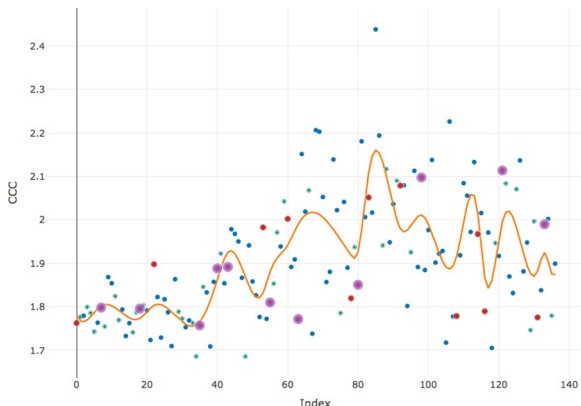

**Fig 3. The distribution of CCC (condition 2) in Russian modelled by spline regression; red dots indicate samples with more than three new words, purple dots—with more than five.**

**Table 2. Adjusted _R_-squared and _p_-value of linear and spline regression models.**

|  | English | | Russian | |
|---|---|---|---|---|
|  | **Linear** | **Spline** | **Linear** | **Spline** |
| $R^2$ | 0.43 | 0.56 | 0.20 | 0.40 |
| $p$ | $< 0.001$ | $< 0.001$ | $< 0.001$ | $< 0.001$ |

cannot be extended by including one more adjacent node [32] is, in our case, a group of words that appear in a given collection of comments only together. For example, we encounter in the first English sample the following maximal clique: [*mixer, save, manage, thing, probably, man, suit, hold, black, crash, van*]. It is maximal because there is no such comment in the whole sample where all these words are used alongside some others.

The concept of maximal clique turns out to be very helpful in explaining the opposite CCC trends in English and Russian. As shown in Table 3, the number of maximal cliques in the first Russian sample is much greater than in the first English sample. In the consecutive samples, this number continuously decreases in Russian and increases in English, which results in two much more comparable numbers in the last samples.

The difference between two languages with regard to the growth of the networks of words in the first sample of comments can be observed by comparing S1 and S2 Figs (English and Russian). The English network is sparse from the beginning, its build-up involves both centre and periphery (hence a small number of maximal cliques), while the Russian network starts from a very dense state: new comments mostly repeat the words lying in the centre of the network (hence a much bigger number of maximal cliques). Later, these two networks develop in opposite directions: while the English one solidifies its centre, the Russian one widens its periphery. Eventually both ways lead to a similar configuration.

More formally, this process can be explained by comparing the change in two measures: mean degree centrality and mean betweenness centrality of a network at each stage of its growing. Degree centrality for a node *n* is usually defined as the fraction of nodes in the graph it is connected to. Betweenness centrality for a node *n* is the sum of the fractions of all pairs of nodes' shortest paths that pass through it [33, 34]. Calculating average number of ties of all nodes in a network and average number of short paths that go though them can give us important insight into how dense/sparse this network is at a particular moment in time. Given the graphs in S1 and S2 Figs, one would expect that, at initial stretch, 1) average degree centrality of the Russian network will be greater than that of the English one while 2) average betweenness centrality of the English network will be greater than that of the Russian one.

Plots in Fig 4 as well as two conducted *t*-tests confirm these expectations. Under the null hypothesis that English and Russian samples have identical average values, I got the following results: 1) for mean degree centrality $t = -3.63$, $p < 0.001$, 2) for mean betweenness centrality $t = 2.57$, $p = 0.01$. This supports my view of the Russian first sample's network as densely built around its centre and of the English first sample's network as more sparse but highly connected, allowing a free flow of information in the graph.

To sum up, Study 1 has shown a remarkable difference in how participants of the experiment described the picture when they were not able to see each other's comments (condition

**Table 3. Number of maximal cliques in English and Russian first and last samples.**

|  | First sample | | Last sample | |
|---|---|---|---|---|
|  | **English** | **Russian** | **English** | **Russian** |
| *Maximal cliques* | 239 | 613 | 510 | 488 |

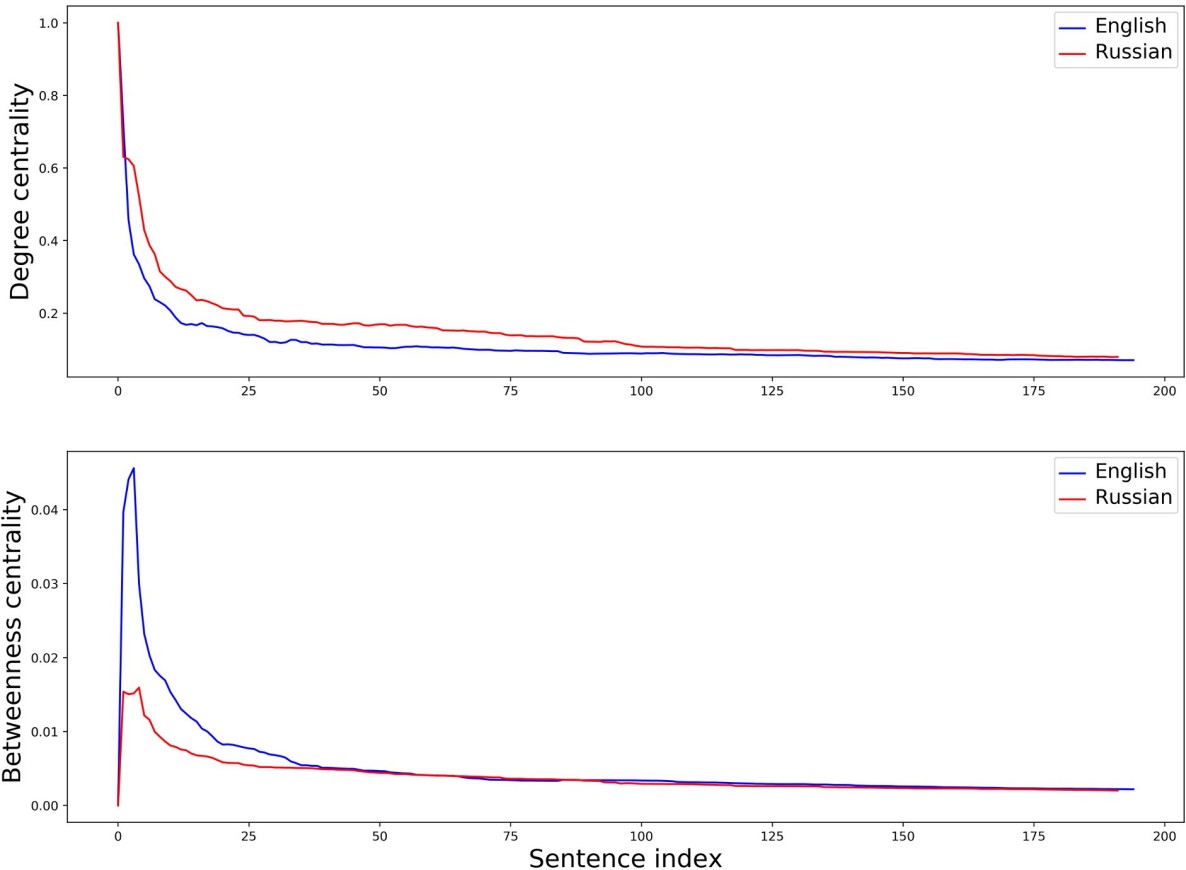

**Fig 4. Average degree and betweenness centrality in English and Russian networks.**

1) and when they were (condition 2). In the former case, their CCC remained unaffected by the growing distance separating the comments. In the latter case, the increasing sparseness of the comments' space resulted in linear change of the CCC value, negative in English and positive in Russian. As I tried to show, both these general trends have some sort of rhythmic structure to them. Specifically, they are comprised of the series of shorter stretches, each of which is negatively correlated with the sample index.

What is observed can be explained in the following way. The mechanism of collective creativity is likely to be the same in both languages: every now and then, at more or less regular intervals, a new batch of words is thrown into the mix of ongoing picture elucidation process. Upon their appearance, these new batches immediately increase the CCC of the sample since they enrich variety of contexts the nuclear words are used in. Later on, these newcomers either get abandoned as gratuitous embellishments or start being repeated themselves, which leads to a continuous decrease of CCC till the latest batch arrives.

As for the general, either upward or downward trend, I contend that it is dependent solely on the starting position. If we convince ourselves that, for any given picture, there is a limit state where all the words that can be sensibly used for the description are used, it is evident that to reach this limit state, new commenters, in their desire to be linguistically creative, will necessarily keep introducing words that have not yet been uttered. The positive or negative character of change in CCC, then, will be predetermined by the structure of the network of words in the first-to-appear comments.

Provided that initial structure is dense, with high average degree centrality (that is, first comments are mostly formulaic), development will go along the line of increasing variability, moving from centre to periphery of the network. Provided that initial structure is sparse, with high average betweenness centrality (that is, first comments are mostly creative, with just a few recurrent nuclear words), development will go along the line of increasing repetitiveness, moving from periphery to centre of the network. The former constitutes the Russian case of growing CCC, the latter is the English case of declining CCC.

The both models of network development are schematised in Fig 5, where red nodes and edges provide initial configurations while blue nodes and edges represent later stages. It can be seen that even for these toy networks, discrepancies in the number of maximal cliques, average degree and betweenness centrality are substantial and in line with my reasoning.

To find out whether the choice of one of those two models depends upon language or some experimental variation that remained unaccounted for, further research is needed.

## Study 2. Data and methods

After introducing the concept of collective language creativity and describing an algorithm to model it, I will now discuss how cognitive priming effects are set off by creative antipriming efforts of the participants in the communication process. To analyse the experimental results appropriately, I, first, identified four fixed zones of picture description (Fig 6) and then divided each English and Russian comment into the groups of lemmatised words pertaining to these

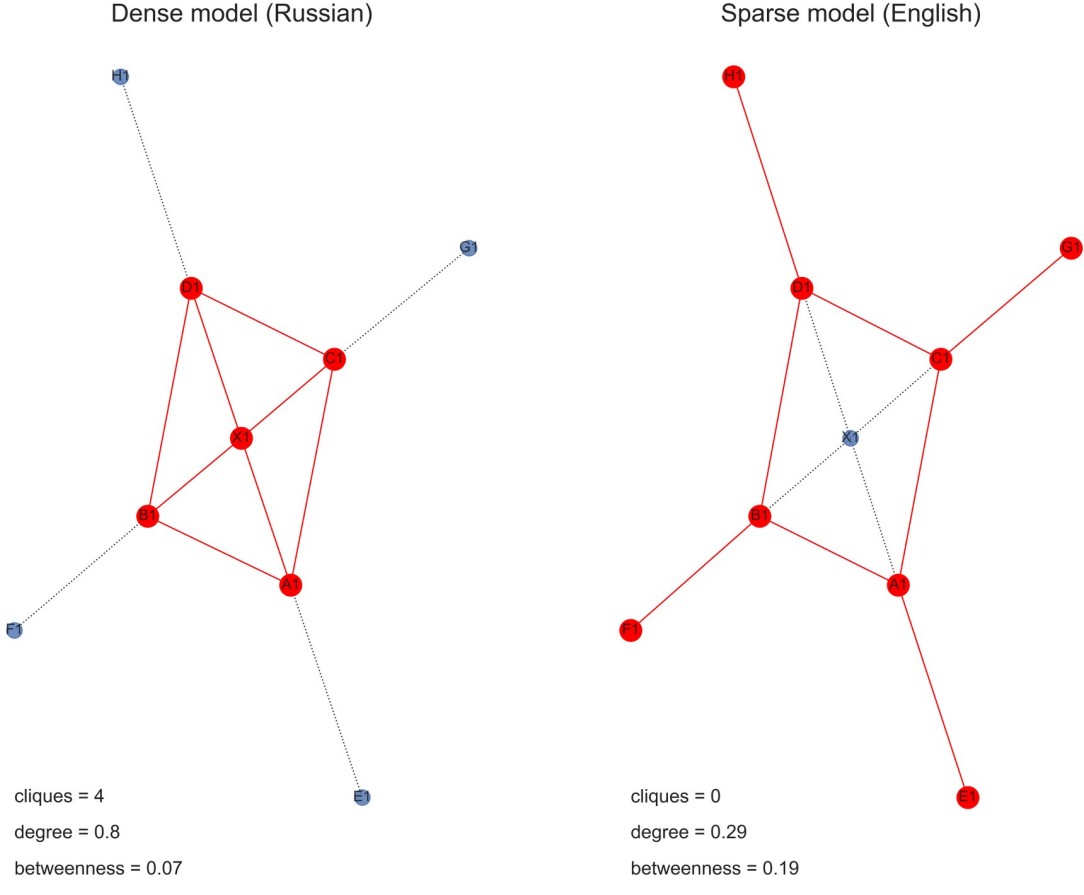

Fig 5. **Two models of network development.**

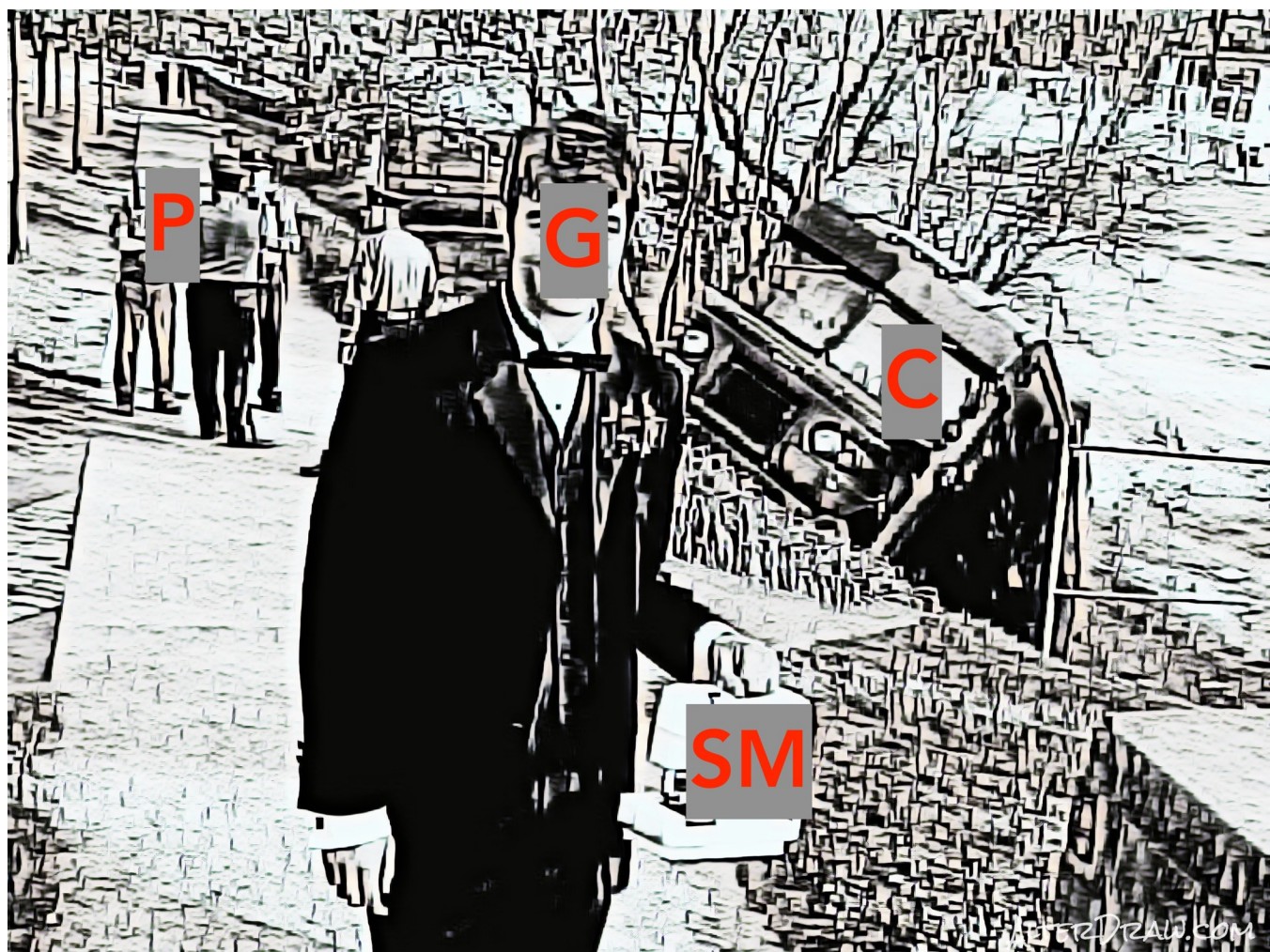

**Fig 6. The (visually modified) picture participants of the experiment were asked to describe with annotated description zones (G for guy, SM for sewing machine, C for car, P for police).**

zones. (Fig 6 is not the original image used in the study, but a similar image provided here for illustrative purposes only.)

Four main description zones were further subdivided into a number of subzones, either three or two, according to the structures in Tables 4–7 (for compactness, only English examples from the first three comments are provided). All words describing the mutual positioning of figures in the picture (like *near*, *behind*, *next to*, *in front of*) were collected in the eleventh column. As a result, I got four tables, two languages in two conditions; each table had 11 columns, one for each subzone plus extra column, and the number of rows corresponding to the

**Table 4. Subzones of zone G.**

|  | **G1 (guy himself)** | **G2 (guy's attire)** | **G3 (guy's actions)** |
|---|---|---|---|
| Comment 1 | *young, man* | *in, black, suit* | *smile* |
| Comment 2 | *groom* | *wear, tuxedo* | *pose, for, photo* |
| Comment 3 | *white, guy* | *nicely, dress* | *stand* |

**Table 5. Subzones of zone SM.**

|  | SM1 (sewing machine itself) | SM2 (way of holding the object) |
|---|---|---|
| Comment 1 | *sewing, machine* | *with* |
| Comment 2 | *white, sewing, machine* | *hold* |
| Comment 3 | *tailoring, machine* | *stay, with* |

number of comments. The subzones were arranged in the following order: G1, G2, G3, SM1, SM2, C1, C2, C3, P1, P2. Naturally, not all comments covered all subzones, such cases were marked in the tables with special sign.

After creating the tables with dissected comments, I turned each comment with the exception of the very first one into a numeric vector of 11 numbers depending on how many words this comment inherited from its immediate predecessor zone-wise. For each zone, the respective number was calculated according to the following algorithm: *length([word for word in subset[i] if word in subset[i-1]]) / length(subset[i])*, that is, as the proportion of words in subzone $S_i$ of the comment $C_n$ that are present in the subzone $S_i$ of the comment $C_{n-1}$. Value of 1 indicated that all words were repeated, value of 0 that no word was repeated. Cases where a comment inherited an empty subzone from its immediate predecessor were assigned -1. For example, given the comment $C_{n-1}$ '*A man in a wedding suit with a mini sewing machine*', the comment $C_n$ '*A young boy in formal suit with sewing machine. And a bus plummeting behind*' had the following vectorised form: [G{0, 0.66, -1}, SM{1, 1}, C{0, 0, 0}, P{-1, -1}, 0].

## Study 2. Results

To better understand the repetition patterns of both languages in both experimental conditions, I created heatmaps of the four tables. Values in these graphical representations of data are depicted by color: warmer colours stand for higher values, colder for smaller ones. The results are plotted in Figs 7 and 8 (English and Russian). The patterns are more distinct in Russian and somewhat blurred in English, however, the trends are similar. Most important among them are, first, the fact that the overall structure is determined by empty subzones (dark blue colour) and second, that with regard to the constellations of empty subzones, comments in the first condition are aligned vertically, while comments in the second condition are aligned vertically and horizontally.

To prove it, one must show that the comments in the first experimental condition in both languages constitute a holistic structure, while comments in the second condition are organised in clusters. That was done by transforming each vector in the data into a vector of -1s and 0s by replacing all its numbers not equal to -1 with zeroes and applying to it two clustering conditions simultaneously. After grouping this vector together with the preceding vectors of a still open cluster, 1) the mean number of this cluster's empty subzones should not decrease, 2) the mean Manhattan distance between each pair of this cluster's adjacent vectors should not increase. In case both these conditions were satisfied, the vector was assigned to the proposed cluster. In case either of them failed, the proposed cluster got closed and the vector was

**Table 6. Subzones of zone C.**

|  | C1 (car itself) | C2 (car's movement) | C3 (car's position) |
|---|---|---|---|
| Comment 1 | *truck* | *in* | *ditch* |
| Comment 2 | *ups, truck* | *slip, in* | *river* |
| Comment 3 | *ups, van* | *go, over* | *embankment* |

**Table 7. Subzones of zone P.**

| | P1 (men themselves) | P2 (men's actions) |
|---|---|---|
| Comment 1 | *group, of, cop* | *converse* |
| Comment 2 | *policeman* | *discuss, the, matter* |
| Comment 3 | *delivery, man, police* | *talk, in, a, circle* |

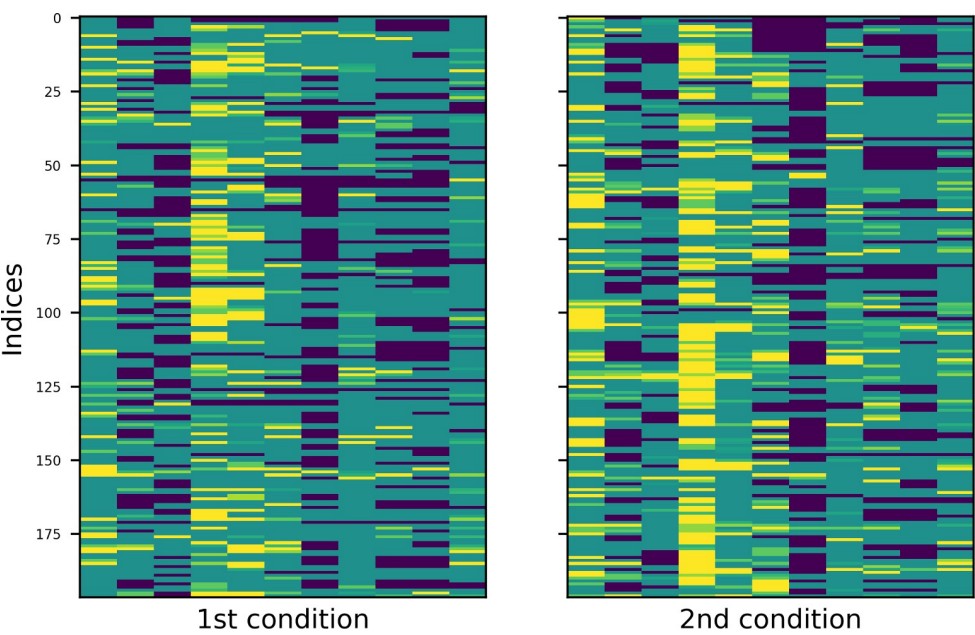

**Fig 7. Heatmap of English data in two experimental conditions.**

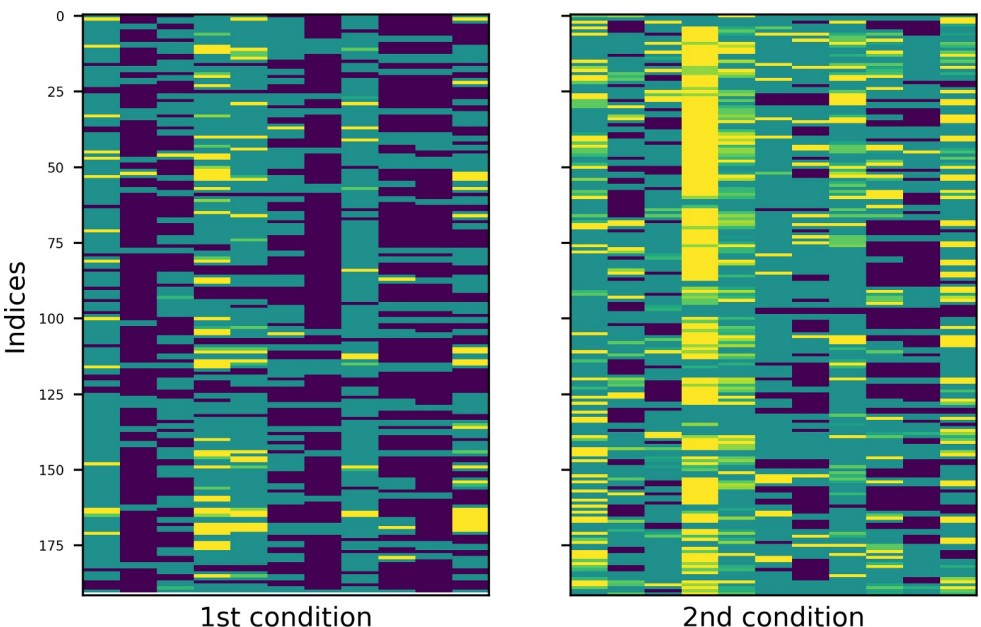

**Fig 8. Heatmap of Russian data in two experimental conditions.**

assigned to a new cluster. More formally, for each pair of adjacent vectors $V_i$, $V_j$, where $j = i+1$ and $\{V_i \ldots V_1\} \in C_k$, if $F_1(\{V_j \ldots V_1\}) > = F_1(\{V_i \ldots V_1\})$ and $F_2(\{V_j \ldots V_1\}) < = F_2(\{V_i \ldots V_1\}) = > V_j \in C_k$, otherwise $V_j \in C_{k+1}$, where:

$$F_1 = \frac{1}{m} \sum_{n=1}^{m} \sum_{r=1}^{11} |x_{n,r} = -1|, \quad F_2 = \frac{1}{m} \sum_{n=1}^{m-1} \sum_{r=1}^{11} |x_{n,r} - x_{n+1,r}|.$$

By implying these two conditions, I made sure that the members of each cluster would be similar not only in the number of empty subzones but also in their positioning. The results of the clustering procedure were conspicuous. In the first experimental condition, no clusters were identified. As for the second condition, with both languages, the number of identified clusters turned out to be 34. The fact that the numbers match is of course just a coincidence, however, the results clearly show that comments written 'together' are indeed structured very differently than those written 'alone'. Analysis of the clusters' boundaries suggests that after every five or six comments description schema gets refreshed by means of 1) shifting the focus of attention to the subzones that earlier remained unheeded and 2) omitting the subzones that were brought into focus by the preceding group of messages.

## Study 2. Discussion

What the results of the comments' cluster analysis tell us is that in the first experimental condition, only vertical lines of force influence the choice of lexical matter, and the positioning of these lines is most probably determined by the degree of visual and conceptual prominence of different subzones in the picture. Thus, the subzones most frequently left empty by both English and Russian participants are: G2, G3, C2, P1, P2.

The group of policemen (P1) is mentioned seldom because it is in the furthest background and does not attract much attention. Naming the actions of the guy (G3) and the police officers (P2), as well as the movements of the car (C2) is rare because the picture is static and induces mostly nominal sentences. Finally, guy's attire (G2) is probably overshadowed by a much more eye-catching detail—white sewing machine in his hands.

With the second experimental condition, the situation is different. All the above mentioned subzones, though of course remaining less visually prominent, get involved in the picture elucidation process much more frequently. It happens, as I contend, due to the participants' desire to be linguistically creative, say something that has not been said by other commenters within a reasonable range of consideration. As a direct consequence, this process results in a checkered pattern of repetitions: to avoid reiteration of the same words in a particular subzone that for a while has been kept formulaic, a new writer introduces a batch of new words in there, but at the same time, he or she might light-heartedly reuse the words in another, neighbouring subzone, where, presumably, the degree of repetitiveness is still tolerable. That is why I say that the comments in the second experimental condition are aligned both vertically and horizontally: the selection of each word is influenced not only by the words in the same subzone of different comments but also by the words in the different subzones of the same comment.

To show that this is really the case, two probabilistic graphical models were constructed, more specifically, two dynamic Bayesian networks (DBN), where the inference is performed upon a network comprised of consecutive time slices *t-1* (immediately preceding comment) and *t* (immediately following comment), so that only the nodes (comments' subzones) in slice *t-1* with children in slice *t* are part of the inference [35, 36]. In Model A (see model's template in Fig 9), I assumed only vertical conditioning of each subzone by its predecessor. In Model B (see model's template in Fig 10), additionally to that, horizontal conditioning by some subzones in the same comment believed to be central for elucidation was modelled. The names of

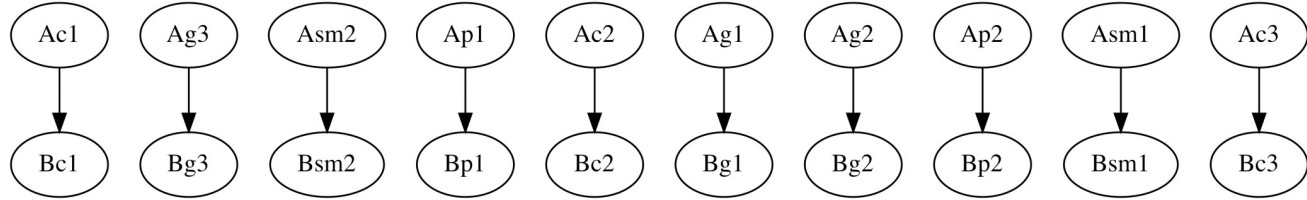

**Fig 9. Model A with vertical (between-comment) conditioning.**

the nodes in the models are written as follows: first letter stands for either time slice *t-1* (A) or time slice *t* (B); the rest of the name is the name of respective subzone (see Tables 4–7). As a rough approximation, I assumed the following set of horizontal dependencies: 1) within zones: G1—{G2, G3}, SM1—{SM2}, C1—{C2, C3}; P1—{P2} and 2) between zones: G1—{SM1, P1, C1}. The eleventh column of the initial tables that contained words describing the mutual positioning of figures in the picture was excluded from both DBNs, as it would be hard to model its independence from other nodes.

The conditional probability distributions for both models were learnt from the data. To make the distributions discrete and thus more manageable, I rearranged the initial tables of comments' vectors (X—> Y) so that for each node, only three value assignments were possible: 1) $Y_i = 1$ if $X_i = \{-1, 1\}$, 2) $Y_i = 2$ if $0 < X_i < 1$, 3) $Y_i = 3$ if $X_i = 0$.

That is to say, I collapsed together 1) the cases where two identical subzones of adjacent comments shared at least one word, but the later-to-appear comment also introduced at least one new word in this subzone (value of 2); 2) the cases where two identical subzones of adjacent comments were either verbatim or both empty (value of 1). For each node $Y_i$, a distribution over its values $P(Y_i | Parents)$ was specified given each possible joint assignment of values to its parents in the model. For a node with no parents, the distribution was conditioned on the empty set of variables, hence, constituting a marginal distribution $P(Y_i)$.

After constructing the models, I used them to recreate the observed repetition patterns for each language and each experimental condition by sampling from the specified distributions. The simulation ran as follows. At the start, vectors corresponding to the chronologically first comments in all four datasets were taken as evidence representing a time slice *t-1*, based on

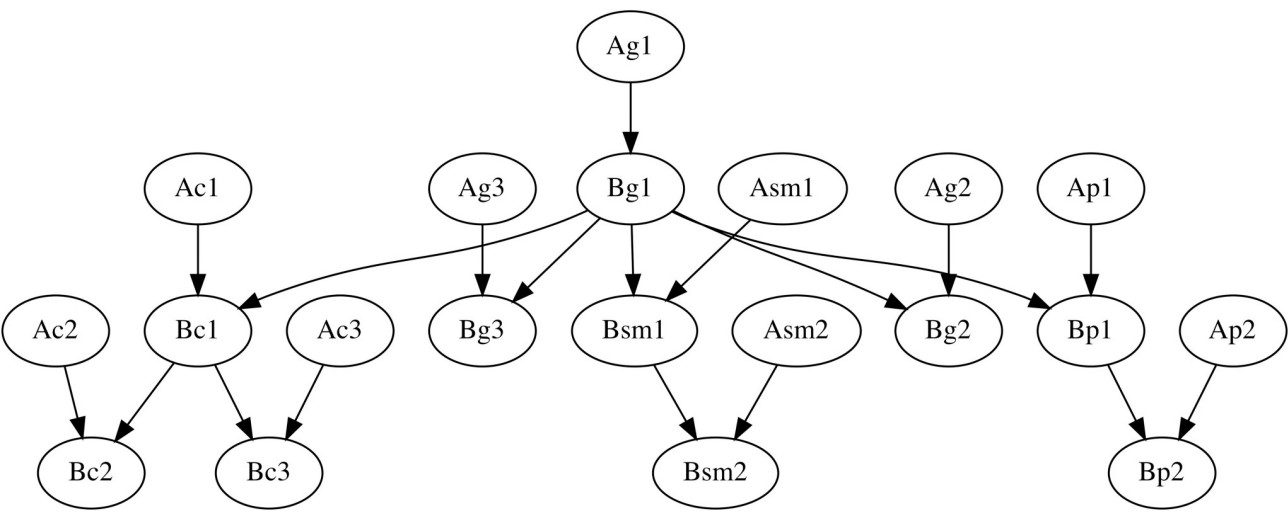

**Fig 10. Model B with vertical (between-comment) and horizontal (within-comment) conditioning.**

which a vector for time slice *t* was obtained as the most probable given the structure of the Bayesian network. At each consecutive step, a vector obtained by sampling was considered to be new evidence at a time slice *t-1*, the A-level variables of the conditional probability distributions were set to respective values, and the following vector at a time slice *t* was simulated. The simulation aborted when the number of vectors in the recreated dataset equaled the number of vectors in the original one.

The heatmaps of English and Russian data are provided in Figs 11 and 12, respectively. Experimental conditions are marked with initial ordinal numbers '1st' ('alone') or '2nd' ('together'), observed data have extension '_obs' in their names; data simulated by models A and B are subscribed accordingly. Now, to show that the repetition patterns of the comments have vertical structure in the first condition and checkered in the second, one must prove that the former are equally well reproduced by models A and B, while for the latter, the model B is more appropriate.

In order to prove it, I wrote an algorithm attempting a series of random walks across each of the 12 datasets, both observed and simulated. The logic of random walk was in each case as follows: 1) a random cell *n* was chosen in a table, 2) a set of possible moves was defined for it, so that this set formed a square of nine cells with *n* in its centre, 3) out of this set, a subset of cells with the same value $v \in \{1, 2, 3\}$ as that of *n* was drawn, 4) out of this subset, the next move *n+1* was randomly selected, 5) for the cell *n+1*, the whole process was repeated. Random walks could proceed in either horizontal direction but were constrained to move only downwards, towards the bottom line of the dataset.

Programmed as this, the random walks highlight the predominant repetition patterns disregarding the exact values of the cells. Including cases of no repetitions (values of 3) in the repetition patterns may sound counterintuitive only at first hearing. It is evident that in a structure, either vertically aligned or checkered, the cells not filled with repeated words will be filled with novel words. Thus, I contend that two datasets are similar (that is, the simulated one is an accurate replica of the observed one) if random walks across them, having started at the same cell, spend similar amount of time in the same columns and rows.

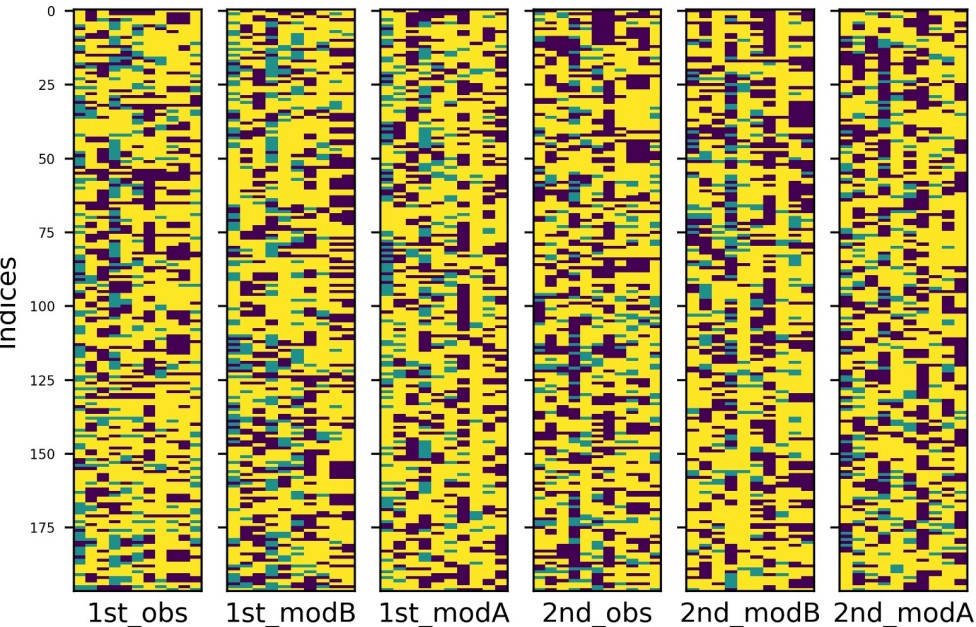

**Fig 11. Heatmaps of observed and simulated repetition patterns (English).**

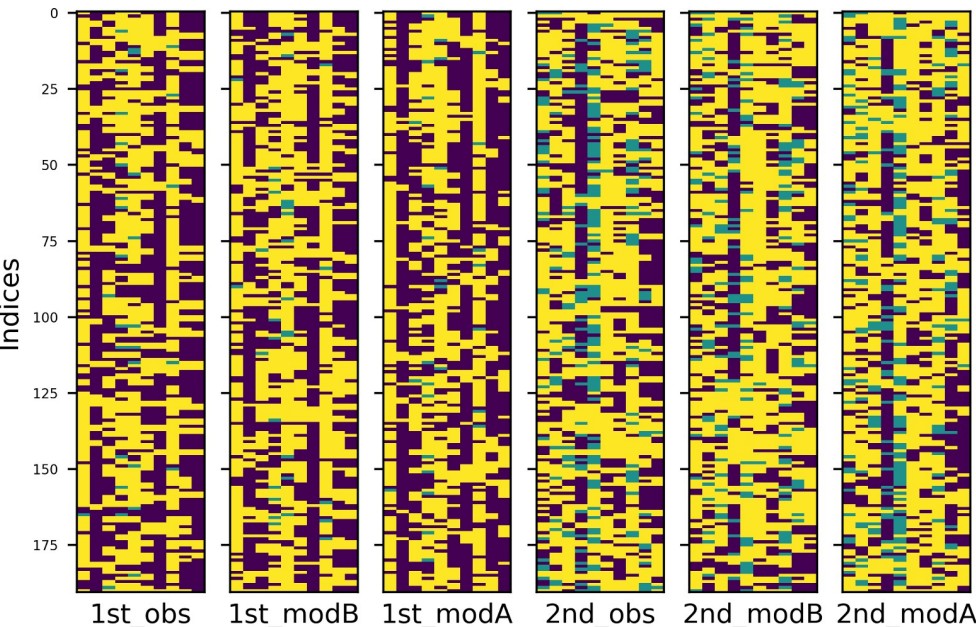

**Fig 12. Heatmaps of observed and simulated repetition patterns (Russian).**

Some examples, one for each dataset, can be found in Figs 13 and 14 (English and Russian). As expected, in both languages, patterns in two experimental conditions differ significantly: in the first one, the random walk algorithm got confined to small number of columns, unrolling vertically, while in the second one, it explored most of the available space, making sweeping diagonal shifts to the left or to the right extremities of the tables. From the provided examples, it is also evident that datasets generated by model A were a better fit for the original datasets in the first condition, while their counterparts generated by model B more accurately replicate the unwinding dynamics of the original datasets in the second condition.

To test this observation formally, for each language-condition pair, I generated 10 datasets using model A and 10 datasets using model B. Then, for each dataset, I conducted a series of 10 random walks, each starting from the first row of a different column in the table. The process was repeated three times for each column, and the results were averaged. For each cell covered by a random walk, its *x*-axis and *y*-axis coordinates were recorded. The loss function was calculated column-wise as follows:

$$LF_{Column} = \frac{1}{3}\frac{1}{m}\sum_{n=1}^{3}\sum_{i=1}^{m}|O_{n,i}^{x} * O_{n,i}^{u} - S_{n,i}^{x} * S_{n,i}^{u}|,$$

where *O* is observed dataset, *S* is sampled dataset, *x* is horizontal coordinate, *u* is vertical coordinate, *n* is the number of trials, and *i* is the number of steps in a random walk.

Statistical analysis of the values of loss function provided in Table 8 justifies my hypothesis: there is indeed no significant difference between first-condition datasets generated by models A and B (English: $t = 1.65$, $p = 0.11$; Russian: $t = 0.91$, $p = 0.37$). As for the second experiment condition, datasets generated by model B are significantly more accurate in replicating original data (English: $t = 2.71$, $p = 0.01$; Russian: $t = 2.13$, $p = 0.04$). Besides, one can note that datasets in the second condition are far more multifarious in terms of repetition patterns than their counterparts in the first condition, as suggested by significantly greater errors.

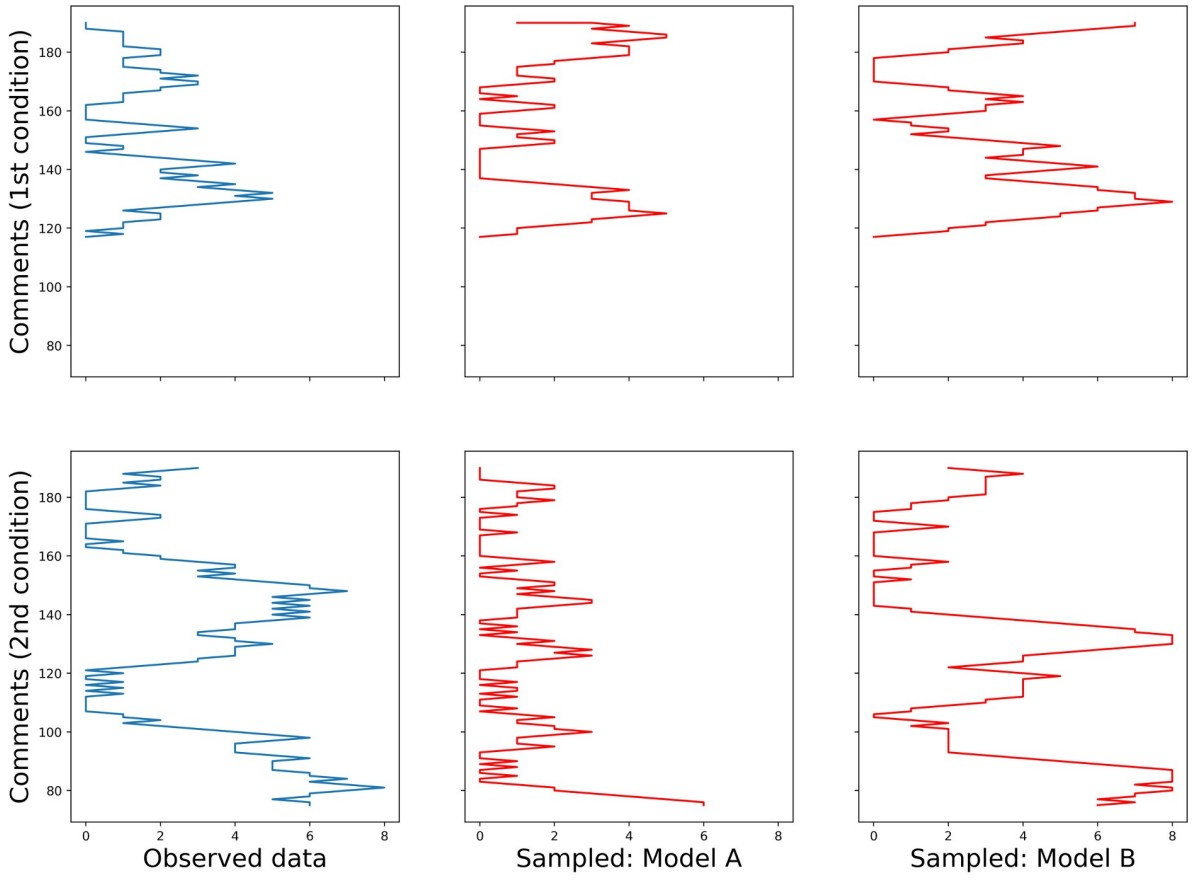

**Fig 13. Examples of random walks across observed and sampled data (English).**

## Conclusion

In this paper, I introduced and discussed the notion of collective language creativity that I understand as a product of interacting of two major factors: cognitive priming effects and creative antipriming efforts of the participants in the communication process. In study 1, it was proven that collective language creativity can be measured by the so-called 'troll coefficient' introduced in the earlier research, that is, as the ratio of the proportion of repeated content words among all content words to the proportion of repeated content word pairs among all content word pairs in a specific sample of messages.

It was shown that in the messages written by people aware of what others have written on a relevant topic before them, a clear linear trend in the development of CCC is observed. Each increase in chronologically aligned sample index predictor led to a decrease in CCC response in English and increase in CCC response in Russian. I contended that this difference in the direction of the linear relationship was dependant solely on high or low base effect which can be measured by means of average degree and betweenness centrality of the network of words in a sample. Provided that initial structure is dense, with high average degree centrality, development will go along the line of increasing CCC. Provided that initial structure is sparse, with high average betweenness centrality, development will go along the line of decreasing CCC.

In study 2, I explored how exactly priming and antipriming effects work together producing collective language creativity. By means of cluster analysis and Bayesian network modelling, it

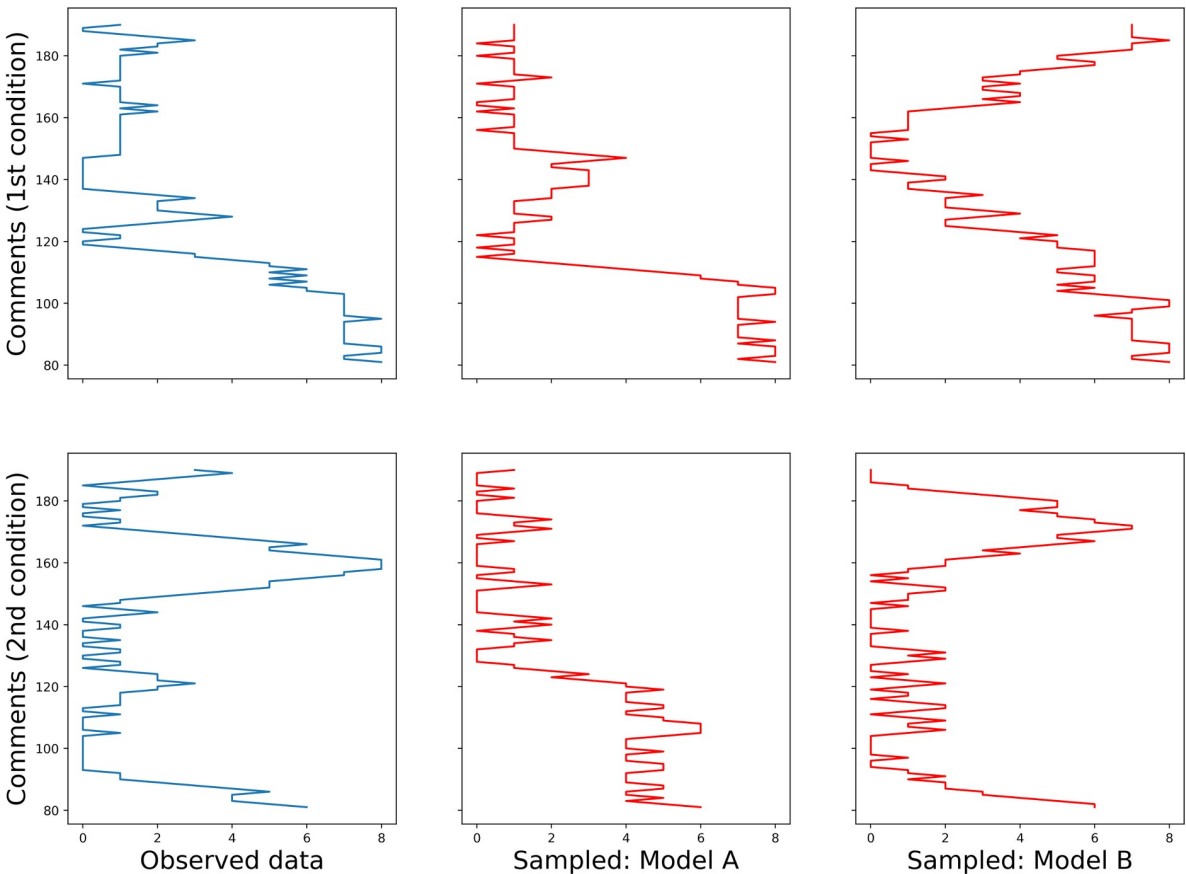

**Fig 14. Examples of random walks across observed and sampled data (Russian).**

was shown that patterns of repetition, for both languages, differ drastically depending on whether participants of the experiment had to communicate their message without being able to see what anyone else has written or had the possibility (but no necessity) of reading what others have posted before them.

**Table 8. Results of comparing random walks across observed and sampled datasets.**

| Column | English | | | | Russian | | | |
|---|---|---|---|---|---|---|---|---|
| | Model A | Model B | Model A | Model B | Model A | Model B | Model A | Model B |
| | 1st con. | 1st con. | 2nd con. | 2nd con. | 1st con. | 1st con. | 2nd con. | 2nd con. |
| G1 | 1552 | 1411 | 2170 | 1654 | 1002 | 909 | 2189 | 2195 |
| G2 | 1718 | 1464 | 2278 | 1884 | 1060 | 845 | 2139 | 2174 |
| G3 | 1662 | 1472 | 1872 | 1837 | 937 | 1021 | 2145 | 2219 |
| SM1 | 1456 | 1365 | 2034 | 1778 | 1031 | 879 | 2257 | 2100 |
| SM2 | 1414 | 1511 | 1867 | 1843 | 1295 | 1172 | 2273 | 2117 |
| C1 | 1455 | 1605 | 2002 | 1740 | 981 | 1261 | 2086 | 2145 |
| C2 | 1541 | 1448 | 1963 | 1782 | 893 | 1067 | 2381 | 1958 |
| C3 | 1603 | 1518 | 1742 | 1797 | 1043 | 957 | 2555 | 2152 |
| P1 | 1498 | 1452 | 2262 | 1935 | 1176 | 1099 | 2152 | 2217 |
| P2 | 1444 | 1471 | 2038 | 2126 | 1315 | 956 | 2339 | 2148 |
| Mean | 1553.9 | 1471.0 | 2022.8 | 1837.6 | 1073.3 | 1016.6 | 2251.6 | 2142.5 |

**Table 9. Example of the predominant repetition pattern in the second experimental condition (shaded are the verbatim repeated chunks).**

|  | G1 | G2 | G3 |
|---|---|---|---|
| Comment 104 | young, man | — | — |
| Comment 105 | young, man | wear, tuxedo | go, to, wedding |
| Comment 106 | man | wear, tuxedo | smile |
| Comment 107 | man | in, black, suit | smile |

In the former case, where no priming or antipriming was present, a vertically organised pattern was observed, influenced only by the visual prominence of different zones of the picture being described and the degree of lexical variability pertaining to the naming of different objects in the picture (compare the numbers of possible names for a young man and for a sewing machine). In the latter case, however, a peculiar checkered pattern of repetitions emerged: to avoid reiteration of the same words in the description of a particular object that for a while has been kept formulaic by other commenters, a new writer would introduce a batch of new words, but at the same time, would repeat his or her predecessors in talking about another object, the degree of description repetitiveness of which is still tolerable. As a result, one gets a pattern provided for illustration in Table 9.

It requires further investigation to see how many consecutive repetitions are needed to trigger such shifts and whether there exists any dependence between susceptibility to priming/antipriming, picture's composition, and the information structure of interacting sentences.

Finally, it is necessary to discuss some limitations of this project. It has already been mentioned in study 1 that the selection of languages was an ad hoc choice. I do not know whether there is something in the way English and Russian are structured or used in writing that could have contributed to the observed difference. It is also hard to predict, for any other given language, which one of the two identified models of words' network development will be preferred.

Second, reading the comments was completely optional for the participants. I did not collect any data about whether (or how much) people actually paid attention to previous comments. It may be the case that the Markovian assumption I made in study 2 is an oversimplification. We can easily convince ourselves that some of the participants, who had to scroll down to the bottom of the page to find their input row, would pay more attention to the first couple of comments and the last couple of comments while some would heed just the few closest in time. These different scenarios can be the source of some variation that, as of yet, remains unaccounted for.

## Supporting information

**S1 Fig. Animated graph of the network of words in the first sample of comments (English).** (MP4)

**S2 Fig. Animated graph of the network of words in the first sample of comments (Russian).** (MP4)

## Author Contributions

**Investigation:** Sergei Monakhov.

**Methodology:** Sergei Monakhov.

**Software:** Sergei Monakhov.

**Visualization:** Sergei Monakhov.

**Writing – original draft:** Sergei Monakhov.

**Writing – review & editing:** Sergei Monakhov.

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
