## [Decision Letter · Decision Letter 0]

26 Aug 2021

PONE-D-21-18394

Collective language creativity as a trade-off between priming and antipriming

PLOS ONE

Dear Dr. Monakhov,

Thank you for submitting your manuscript to PLOS ONE. First of all, apologies again for the delay in reaching a decision on this submission. As I previously indicated, I have had difficulties in securing an appropriate combination of reviewers. Following discussion with a senior editor, we have agreed in this instance that we are able to proceed on the basis of a single external review, combined with my own opinion of your work. Both I and the external reviewer feel that your manuscript has merit but does not fully meet PLOS ONE’s publication criteria as it currently stands. Therefore, we invite you to submit a revised version of the manuscript that addresses the points raised during the review process.

The external reviewer, who is an expert in priming, finds your manuscript clearly written and the study well-designed, as do I. They identify a number of points that need clarifying - having also read the manuscript closely I agree with all the points raised and recommend that you address them with suitable changes to the manuscript. This review noted that a couple of figures could be removed. Whilst I tend to prefer illustrative figures to reams of text, I did wonder if perhaps there was scope to rationalise the number of figures slightly, taking care to include only those that make a new point.

I have some additional comments that I think mostly amplify the recommendations made by the reviewer. Please could you also address these in a revision:

Definition of the creativity measure. I was unsure as to what was meant by the number of repeated words. For example, would the string “A B A B” have two repeated types (A and B) or four repeated tokens (A twice, B twice). Similarly with pairs: by pair do you mean consecutive words, or all pairs of words? If the former, do they overlap; if the latter does order matter? Again, taking the above string, I could construe the following as possible sets of pairs: {AB,AB} {AB,BA,AB}, {AB,AA,AB,BA,BB,AB}, {AB,BA,AA,AB,BA,BA,AB,BB,AB,BA} and possibly other examples. Again, the meaning of repeated also needs to be specified, e.g, in the case {AB,BA,AB} they may be either two or three repeated pairs, depending on whether we are counting types and tokens. I think it would help to provide some short sample texts with the complexity measure calculated, perhaps even examples from the experiments, so the reader is absolutely sure as to how this measure is calculated. I think this is important because the measure features so heavily in the analysis. (Actually, re-reading the manuscript I think this is a between-text rather than a within-text measure, which certainly needs to be clarified)Related, I was unsure as to why, in line 212, q=1 if there are no repeated words. My interpretation is that w=p=0 in that case, which would lead to undefined q.Similarly, at the end of Study 1, it would help to provide concrete examples of the transformation of comments into vectors to make this more concrete.The formulae on lines 445 and 568 have dotted square boxes that show up in various places. I think these should simply not be there, rather than represent missing symbols, but these need to be cleaned up.I felt that the conclusions were perhaps limited in scope, mostly recapitulating the specific findings of the study. I think, given the framing in terms of a general conflict between conformity/priming and creativity in the introduction, it would be natural to discuss implications within this context in the conclusion.

My preference is to avoid a second round of review wherever possible. This is generally achieved if authors address all the issues raised through changes to the manuscript as opposed to detailed private replies to the reviewers, although of course it is important to clearly indicate in a response what changes have been made. Furthermore, if you feel very strongly that a specific recommendation is inappropriate, I will of course consider reasoned arguments to this effect (although this may necessitate a second round of review).

We look forward to receiving your revised manuscript.

Kind regards,

Richard A Blythe

Academic Editor

PLOS ONE

Journal Requirements:

2. For this single-authored manuscript, please replace "we" with "I".

3. We note that Figure 1 and 8 includes an image of a participant in the study. 

Please respond by return e-mail with an amended manuscript. We can upload this to your submission on your behalf.

If you are unable to obtain consent from the subject of the photograph, please either instruct us to remove the figure or supply a replacement figure by return e-mail for which you hold the relevant copyright permissions and subject consents. In some cases, you may need to specify in the text that the image used in the figure is not the original image used in the study, but a similar image used for illustrative purposes only. We can make any changes on your behalf.

Reviewers' comments:

Reviewer's Responses to Questions

**Comments to the Author**

1. Is the manuscript technically sound, and do the data support the conclusions?

Reviewer #1: Yes

2. Has the statistical analysis been performed appropriately and rigorously? 

Reviewer #1: Yes

3. Have the authors made all data underlying the findings in their manuscript fully available?

Reviewer #1: Yes

4. Is the manuscript presented in an intelligible fashion and written in standard English?

Reviewer #1: Yes

5. Review Comments to the Author

Reviewer #1: The aim of the study presented in this manuscript was to explain the mechanisms between priming, a cognitive process that activates recently experienced linguistic forms and makes them more likely to be produced, and language creativity, the production of words and phrases that are novel to a context or dialogue. The authors point out that there is an inherent conflict between these two processes that has not been investigated. They aimed to fill this gap by conducting studies in two languages, English and Russian, and presenting a model that seeks to explain the experimental results.

I think the study is clear and well-designed, the data is explored thoroughly and the findings in the paper could definitely benefit the wider field. However, there are some issues that should be addressed before publishing, and some changes in presentation could improve the paper. Most of these involve clarification.

• Line 47: There is no reason for creating the acronym CALU for ‘creative aspect of language use’, it is only used twice in the whole manuscript.

• Line 73: The author poses the question of why priming and creative language use don’t cancel each other out? I’m not sure what this would look like in practice? After all, people have only two choices, either repeat language forms used previously by their interlocutors or not. What is meant by the processes cancelling each other out?

• The paragraph beginning at line 83 should be clarified, perhaps by providing some concrete examples of troll language. From the Sherlock Holmes example, it’s difficult to understand what the author means by an “anomalous distribution of repeated words” characterising the language of internet trolls. The connections aren’t obvious.

• Line 95, first sentence: The author refers to a “theory”, but it isn’t clear what theory is being referenced. It would really help the reader if the theory with regards to “troll-like” comments or reviews were explained more explicitly. It’s also not conceptually clear why a certain number of repetitions should characterise only troll speech. Calling it a “troll effect”, especially if it’s also true to neutral comments that people produce when exposed to other people’s comments seems inaccurate.

• Was there a reason to conduct the study in both English and Russian? If it’s only to see if any effect would be true cross-linguistically, this should be made clear. It’s not obvious if we should be expecting different effects in the two languages in terms of the initial hypothesis. A lot of effort is made by the author to try to explain the differences in the findings between English and Russian, and minimise them, but I wasn’t sure if there was something in the way English and Russian is structured or used in writing that could have contributed to the difference.

• Line 135: It’s not helpful to co-opt a word that’s generally used in a different context in the literature. Using ‘antipriming’ for the absence of priming is a bit confusing and misleading. The author seems to use it interchangeably with creative language use. Why not just stick with the latter?

• The samples for analysis in the first study were created by taking an ever increasing range of comments and choosing 50 of them randomly, starting with 50. This should mean that when the range is small, each random selection has a lesser effect on the data than when the range is bigger. Could this be responsible for the larger fluctuation at higher sample indices? Calculating the CCC for a number of randomisations and taking an average might be a better reflection of the true variability of the CCC as the sample index grows.

• Paragraph starting at line 298: It would be very helpful if the author could provide at least one example of how the findings on maximal cliques relate to the data. What does a maximal clique look like in English? If there are fewer maximal cliques in English at the outset, does that mean that words are less likely to be specific to a particular context and more likely to be reused across the comment? I’m trying to figure out what the data show about how the two languages may operate in the comments, but I’m having a hard time.

• Reading the comments was completely optional for the participants. Did the author collect any data about whether (or how much) people actually paid attention to previous comments? If so, was there individual variation in this? If participants have to scroll down to the bottom to find their input row, would they pay more attention to the first couple of comments and the last couple of comments? Or just the few closest in time to their own comments? Would the author expect this to have an influence on the outcome? I know this isn’t crucial to understanding the findings, but I wonder if it might have some explanatory power. Or if it could be the source of some of the variation?

• Line 471: A “the” is missing before “situation”

• Line 604: “of” should be “or”

• Sometimes the alone condition is referred to as Alone and sometimes as Condition 1 on the graphs. Similarly, the other condition can be Together or Condition 2. It would be good to standardise this.

• I’m not sure Figures 11 & 12 are necessary to include. There is nothing in the left-hand panels and the right hand panels are also not very informative and could probably be described in the text easily. This is optional, but I don’t think they add much to the visualisation of the data, unlike all the other graphs.

6. PLOS authors have the option to publish the peer review history of their article (what does this mean?). If published, this will include your full peer review and any attached files.

Reviewer #1: No

---

## [Author Response · Author response to Decision Letter 0]

5 Oct 2021

Dear editors and reviewers, please accept my sincere gratitude for all your comments and suggestions. They helped to improve the paper a lot! Please find my responses in the attached rebuttal letter.

---

## [Editor Report · Decision Letter 1]

18 Oct 2021

Collective language creativity as a trade-off between priming and antipriming

PONE-D-21-18394R1

Dear Dr. Monakhov,

We’re pleased to inform you that your manuscript has been judged scientifically suitable for publication and will be formally accepted for publication once it meets all outstanding technical requirements.

Kind regards,

Richard A Blythe

Academic Editor

PLOS ONE
---

## [Editor Report · Acceptance letter]

22 Oct 2021

PONE-D-21-18394R1 

Collective language creativity as a trade-off between priming and antipriming 

Dear Dr. Monakhov:

I'm pleased to inform you that your manuscript has been deemed suitable for publication in PLOS ONE. Congratulations! Your manuscript is now with our production department. 

Kind regards, 

on behalf of

Prof. Richard A Blythe 

Academic Editor

PLOS ONE